# Differential phase-diversity electrooptic modulator for cancellation of fiber dispersion and laser noise

Ehsan Ordouie [1,5], Tianwei Jiang [2,3,5] ✉, Tingyi Zhou[2], Farzaneh A. Juneghani[1], Mahdi Eshaghi [1], Milad G. Vazimali[1], Sasan Fathpour [1,4] ✉ & Bahram Jalali[2]

Bandwidth and noise are fundamental considerations in all communication and signal processing systems. The group-velocity dispersion of optical fibers creates nulls in their frequency response, limiting the bandwidth and hence the temporal response of communication and signal processing systems. Intensity noise is often the dominant optical noise source for semiconductor lasers in data communication. In this paper, we propose and demonstrate a class of electrooptic modulators that is capable of mitigating both of these problems. The modulator, fabricated in thin-film lithium niobate, simultaneously achieves phase diversity and differential operations. The former compensates for the fiber's dispersion penalty, while the latter overcomes intensity noise and other common mode fluctuations. Applications of the so-called four-phase electrooptic modulator in time-stretch data acquisition and in optical communication are demonstrated.

Carrying torrents of data between internet hubs, and connecting servers, storage elements, and switches inside data centers, optical fiber communication is the backbone on which the digital world is built. The basic constituents of such links, the optical fiber, semiconductor laser, optical modulator, and photoreceiver, all place limits on the bandwidth and the accuracy of data transmission.

The three most fundamental limitations in optical communication are those placed by the dispersion of the fiber, the laser noise, and fiber nonlinearities[1]. In the transmission of analog signals, the linearity of the electrooptic (EO) modulator is also paramount. In this paper, we propose and demonstrate an electrooptic modulator (EOM) that addresses two of these problems, namely fiber dispersion and laser noise. Specifically, the modulator eliminates the dispersion penalty and the common mode noises, such as the relative intensity noise (RIN), by providing multiple diverse outputs that are processed via simple digital processing. Chromatic dispersion of optical fibers leads to group-velocity dispersion (GVD), which causes optical pulses to broaden in the time domain, and this leads to intersymbol

interference. This places a limit on the maximum data rate that can be transmitted for a given fiber length[1]. Dispersion can be mitigated using optical dispersion compensation, electronic equalization, or a combination of both. The main noise mechanism in semiconductor lasers is spontaneous emission with random phase contribution, leading to RIN and degraded signal-to-noise ratio (SNR) at the receiver side.

The main figures of merit for any optical communication or sensing system are bandwidth and sensitivity. Notwithstanding the speed limitations of the transmitter and the receiver, the bandwidth is primarily constrained by the frequency fading effect due to the dispersion penalty. In a typical optical link or time-stretch instrument, the sensitivity is limited by the laser RIN or the thermal noise of the receiver. With respect to dispersion penalty, there are two main techniques to mitigate it, namely single-sideband modulation (SSB) and phase diversity[2,3]. The SSB technique is difficult to implement in practice, as it is highly sensitive to mismatches in the signal paths in the optical hybrid. Meanwhile, to mitigate the RIN, the differential push-pull modulation can be employed[4]. The main objective of the present

[1]CREOL, The College of Optics and Photonics, University of Central Florida, Orlando, FL, USA. [2]Electrical and Computer Engineering Department, University of California, Los Angeles, Los Angeles, CA, USA. [3]State Key Laboratory of Information Photonics and Optical Communications, Beijing University of Posts and Telecommunications, Beijing, China. [4]Department of Electrical and Computer Engineering, University of Central Florida, Orlando, FL, USA. [5]These authors contributed equally: Ehsan Ordouie, Tianwei Jiang. ✉e-mail: jtw@bupt.edu.cn; fathpour@creol.ucf.edu

work is to create a modulator that is capable of providing both phase diversity as well as differential modulation, concurrently. Existing EOM structures are incapable of simultaneously achieving phase diversity and differential functionalities. Both require a dual output design, but phase diversity is traditionally based on a single electrode, whereas differential operation requires a dual-electrode design.

Optical modulators are key devices in any communication or signal-processing system. Phase and amplitude modulations are two fundamental mechanisms to attain such devices. The Pockels effect (e.g., in lithium niobate (LiNbO$_3$, LN)[5] and polymers[6]) and the free-carrier plasma effect (mainly in silicon[7]) are commonly employed for phase modulation, while the quantum-confined Stark effect (e.g., in silicon-germanium[8] and compound semiconductors[9–12]) is typically used for amplitude modulation. Mach-Zehnder interferometers (MZIs) and microrings[13–15] are usually exploited to convert phase modulation into intensity modulation. Significant advancements in the performance of optical modulators across various material platforms and structures have been reported in recent years. For instance, modulation bandwidths within the subterahertz range are reported in thin-film lithium niobate (TFLN) MZIs[16,17]. Additionally, silicon modulators have shown impressive progress, with reported data rates exceeding 100 Gbit s$^{-1}$[18–20] and modulation efficiencies of below 1 V cm[21,22]. As discussed later, TFLN is the chosen platform for this work. In the following, further deliberation on the shortcomings of existing EOMs based on MZIs is presented.

Advanced coherent communication links rely on modulation formats such as quadrature phase shift keying (QPSK). As shown in Fig. 1a, a QPSK optical modulator consists of two nested MZIs, followed by a phase modulator. This achieves the desired $\pi/2$ phase difference between the in-phase and quadrature components of the optical signal. The dual-polarization (DP) variation of a QPSK modulator is capable of taking advantage of two orthogonal guided modes in optical fibers, albeit with a more complex optical architecture (Fig. 1b).

Neither of these modulators offers phase diversity to cancel out the fiber dispersion penalty. They also do not provide differential modulation for the cancellation of common mode noise. Herein, we propose a unique modulator architecture, dubbed four-phase electrooptic modulator (FEOM), that performs both functionalities at the same time. We demonstrated such a modulator fabricated in TFLN offering a small footprint.

A conceptual FEOM is depicted in Fig. 1c and consists of two single-drive, dual-output Mach-Zehnder modulators (MZMs) nested in another MZM. The four outputs of in-phase ($I$), out of phase ($\bar{I}$),

quadrature ($Q$), and inverse quadrature ($\bar{Q}$) components are also shown. The modulator imparts a $\pi$ phase difference between $I$ and $\bar{I}$ (similarly $Q$ and $\bar{Q}$ components), enabling the attainment of differential operation. Subsequently, the FEOM initiates a $\pi/2$ phase difference between the $\{I,\bar{I}\}$ and $\{Q,\bar{Q}\}$ component sets, which facilitates the realization of phase diversity operation, as shown later in Fig. 2. Furthermore, the two modulators work at the same quadrature point. It should be noted that the terms *in-phase* and *quadrature* have different definitions in FEOM and QPSK modulators. In FEOMs, they are associated with two of the four output components, whereas in QPSK modulators, they refer to two independent inputs to the sub-modulators. The FEOM has only one input for encoding data, which appears to reduce its transmitted bit rate by half compared to a QPSK modulator for identical baud rates. Nonetheless, the ability of FEOMs to remove dispersion-induced nulls in frequency response results in a significantly higher effective bandwidth and hence higher bit rate. In addition, an FEOM is able to cancel the common laser noise and improve SNR, thus much lower bit error rate (BER) compared to QPSK modulators. As depicted in Fig. 1d, and discussed further in the Results section, the polarization of guided modes can be exploited to modify the FEOM architecture to attain only one output channel for telecommunication applications.

Given the use of multiple nested interferometers in the same device, an FEOM is best implemented in an integrated-optic platform and ideally, one that provides a pure EO effect (as opposed to electroabsorption or a combination of both). TFLN is an ideal platform to realize such a circuit. LN is a widely known material for its strong electro- and nonlinear-optic properties. The invention of TFLN on silicon substrates[23] has been a significant breakthrough in photonic integrated circuits (PICs), achieving several milestones[5]. The optical waveguides on this maturing thin-film technology offer unrivaled properties compared with the traditional titanium-diffused or proton-exchanged waveguides. These include high refractive-index contrast waveguides that lead to submicron cross-sections and small bending radii, as well as EOMs exhibiting low-voltage operation, high-speed modulation, or both[16,17,24–36]. The FEOM presented in Fig. 1c is designed and fabricated on TFLN in this work.

We demonstrate the utility of the modulator in two application domains. First, the fabricated FEOM is utilized in an experimental time-stretch system. Second, the utility of this modulator in canceling the dispersion penalty in an optical communication link is demonstrated via simulations.

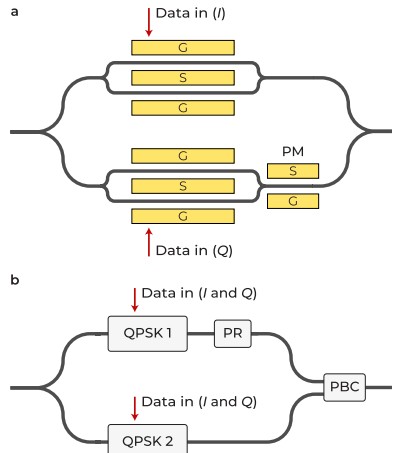

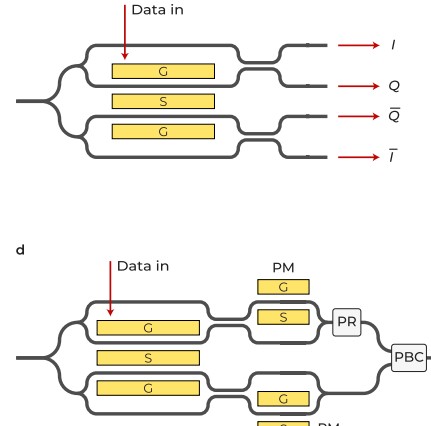

**Fig. 1 | Several common advanced modulation formats and four-phase electrooptic modulator (FEOM) variations: an illustrative example. a** Quadrature phase shift keying (QPSK) modulator. PM phase modulator, G ground, S signal. **b** Dual-polarization QPSK modulator. PR polarization rotator, PBC polarization beam combiner. **c** Proposed and demonstrated FEOM. **d** Dual-polarization version of FEOM.

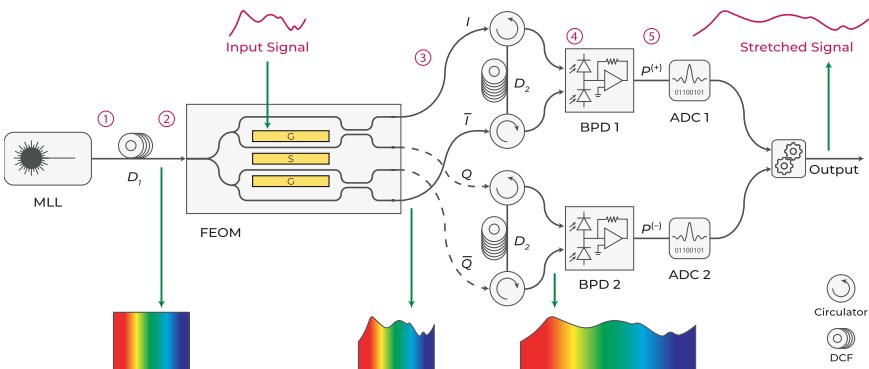

**Fig. 2 | Four-phase electrooptic modulator (FEOM) in a time-stretch system.** Schematic of an FEOM employed in a photonic time-stretch system for demonstrating its phase diversity capability. MLL mode-locked laser, DCF dispersion compensating fiber, BPD balanced photodetector, ADC analog-to-digital converter.

Photonic time stretch is a real-time data acquisition technology[37,38] that has spawned a vast number of scientific and technological advancements[39,40]. This class of real-time measurement systems has been exceptionally successful in capturing single-shot phenomena such as optical rogue waves[41], relativistic electron dynamics[42–44], chemical transients in combustion[45], shock waves[46], internal dynamics of soliton molecules[47], birth of laser mode-locking[48], and single-shot spectroscopy of chemical bonds[49,50]. They have also evolved into high throughput microscopy[51] of biological cells[52], label-free classification of cells[53–55], gyroscopes[56], mid-infrared spectroscopy[57], and many other applications[58–63]. This paper shows the efficacy of the modulator in canceling the dispersion penalty in optical fibers with emphasis on time-stretch systems.

In a photonic time-stretch system, an electrical signal is fed to an EOM in order to modulate a chirped optical pulse. Then, the signal is stretched in the time domain by passing the modulated optical wave through a dispersive element. Eventually, the stretched signal, with a considerably lower analog bandwidth, is converted back to the electrical domain for digitization by using a photodetector. Similar to the mentioned generic optical communication links, bandwidth, and dynamic range are critical figures of merits for photonic time stretch analog-to-digital converters (TSADCs)[64]. In this application, the non-uniform envelope of chirped pulses exacerbates the noise issue. Therefore, simultaneous phase-diversity and differential operations can play a critical impact in improving the performance of time-stretch systems.

This work introduces such a TSADC configuration by utilizing the discussed FEOM architecture. As discussed before, an FEOM has two nested single-electrode dual output EOMs (Fig. 1c) for concurrent operation of differential and phase diversity operations, which cannot be achieved by conventional architectures or off-the-shelf optical components.

## Results

### Theoretical description of the Four-phase ElectoOptic Modulator (FEOM)

To gain a more in-depth understanding of the operational dynamics of the FEOM, analyzing it within the framework of a time-stretch system can be beneficial. We have developed a comprehensive analytical model for the operation of the FEOM in the Methods section.

As depicted in Fig. 2, a broadband optical pulse is first subjected to prestretching via utilization of a fiber-based dispersion element, prior to being introduced into the FEOM. A radio-frequency (RF) signal is added to a pre-stretched optical pulse in an FEOM, resulting in the generation of components $I$, $\bar{I}$, $Q$, and $\bar{Q}$. The $\{I,\bar{I}\}$ set is sent through a pair of optical circulators to a second dispersive fiber, where they are then converted into an electrical signal by a balanced photodetector (BPD). Similarly, the $\{Q,\bar{Q}\}$ set is sent through a pair of circulators to a

third dispersive fiber and detected by a second BPD. The photocurrents prior to the differential operation is given by

$$P_4^{(k)}(t) = P_{\text{env}}(t)[1 + A(t; \delta(k)) + B(t)], \qquad k \in \{I, Q, \bar{Q}, \bar{I}\} \tag{1}$$

where for each component $k$ is the induced phase and equals $\delta = [\pi/4, 3\pi/4, -\pi/4, -3\pi/4]$, respectively. The time-dependent functions $A(t; \delta) = (m/\sqrt{2})\cos(\omega_{\text{RF}}t/S)\cos(\phi_{\text{DIP}} - \delta)$ and $B(t) = (m^2/8)\cos^2(\omega_{\text{RF}}t/S)$. Here, $P_{\text{env}}$ is the photocurrent in the absence of an electrical field, $\omega_{\text{RF}}$ is the angular frequency of the original electric signal, $S$ is the time stretch factor, $\phi_{\text{DIP}}$ is dispersion phase, and $m$ is the modulation index. Equation (1) illustrates the differential functionality of the FEOM, where a $\pi$ phase difference between the $I$ and $\bar{I}$, and similarly between the $Q$ and $\bar{Q}$, components effectively eliminates intensity noise and enhances the system's dynamic range.

In the BPD, after performing the differential operation, the photocurrents are $P_5^{(+)}(t) = P_4^{(I)}(t) - P_4^{(\bar{I})}(t)$ and $P_5^{(-)}(t) = P_4^{(Q)}(t) - P_4^{(\bar{Q})}(t)$, which are equivalent to

$$P_5^{(\pm)}(t) = \sqrt{2}\, m\, P_{\text{env}}(t) \cos(\omega_{\text{RF}}t/S) \sin(\phi_{\text{DIP}} \pm \pi/4). \tag{2}$$

For simplicity, we use $P^{(\pm)}$ instead of $P_5^{(\pm)}$ throughout the remainder of the paper. Due to the differential modulation, the supercontinuum pulse's envelope and the second-order modulation component can be effectively canceled. According to equation (2), the output signals possess a frequency of $\omega_{\text{RF}}/S$, which shows that the signals have stretched in time. The inclusion of the phase term, $\sin(\phi_{\text{DIP}})$, leads to frequency fading, also known as dispersion penalty. This is caused by the destructive interference of RF components generated by the beating of the carrier and the modulation sidebands inside the BPD. The FEOM architecture, as represented in equation (2), manifests a unique and powerful feature in the form of complementary fading characteristics between the $P^{(+)}$ and $P^{(-)}$ channels. These phase-diverse outputs enable the effective counteraction of the detrimental effects of dispersion penalty on the full recovery of the original analog signal through the utilization of the maximal ratio combining (MRC) algorithm[2]. The algorithm increases the SNR as opposed to simply combining the two outputs[3].

### Device design

The devices are optimized for high EO bandwidth and modulation efficiency, and designed for transverse-electric (TE) single-mode operation utilizing the RF module of the COMSOL™ simulation tool. Additionally, the Ansys Lumerical finite-difference time-domain (FDTD) simulation software package is employed for the design of passive components operating at the 1560 nm optical wavelength. A schematic of the FEOM, RF electric field and optical mode profiles, as well as images of a fabricated device, are presented in Fig. 3. The actual

**Fig. 3 | Integrated thin-film lithium niobate (TFLN) four-phase electrooptic modulator (FEOM). a** Schematic of the FEOM. **b** Electric field distribution of overlapping microwave and optical modes of a Mach-Zehnder arm. The TFLN thickness ($t_{LN}$) is 400 nm. The rib waveguide's width and height are 1.3 μm and 110 nm, respectively. The thickness of coplanar waveguide (CPW) electrodes ($t_{Au}$) are 1.0 μm with a signal-to-ground gap ($g$) of 5.5 μm. The thickness of the silicon oxide passivation layer ($t_{ox}$) is 500 nm. **c, d** False-color scanning electron microscope images of the fabricated 3-dB Y-junction and 3-dB directional coupler. Scale bars indicate 4 μm and 1 μm, respectively. **e** Microscope image of a section of a CPW and phase modulators. Scale bar denotes 100 μm.

implemented device (Fig. 3a) incorporates two phase modulators, as opposed to the original FEOM concept (Fig. 1c). The phase modulators allow fine-tuning of the phase, and can neutralize fabrication imperfections and associated deviations in the expected phase of each channel. The layer structure of one of the inner arms of the nested MZMs is illustrated in Fig. 3b with dimensions specified in the caption. A false-color scanning-electron microscope (SEM) and optical microscope images from different sections of the fabricated devices are shown in Fig. 3c–e.

The RF coplanar waveguides (CPWs) are oriented along the *y* axis of the X-cut TFLN crystal in order to capitalize on the highest EO coefficient, $r_{33}$, of LN for TE modes. As shown in Fig. 3a, e bends are added at both ends of the electrodes to facilitate probing and measurement. The bent regions of the electrodes can lead to metallic loss in the optical waveguide underneath. This limitation is mitigated by a thin silicon dioxide buffer layer but comes at the expense of additional processing steps. Moreover, the variation in electrode thickness between the straight and bent sections could result in impedance mismatch and RF reflection at the junction. These factors, along with the RF bending loss, pose potential challenges to the modulator's bandwidth. Despite these hurdles, the achieved EO bandwidth (presented in the next section) meets the goal of observing the first few nulls of the dispersive fiber frequency response.

A comprehensive series of simulations were carried out utilizing a TSADC system as the platform for examination to evaluate the functionalities devised in the design of the FEOM. The parameters of the simulation were chosen to be consistent with the experimental setup described in the study, including the dispersion parameters of the first and second fiber elements, represented by $D_1$ and $D_2$, which were set to −120 and −984 ps nm$^{-1}$ km$^{-1}$, respectively. This resulted in a system stretch factor of $S = 9.2$. The analog-to-digital converter utilized in the simulation had a sampling rate of 50 GSa s$^{-1}$, and an effective number of bits (ENOB) of 7.

The time- and frequency-domain simulation results are displayed in Fig. 4a–f. Figure 4a, d illustrates the input pulse and the distorted output pulse of a time-stretch system. The nulls in the frequency spectrum of the received pulsed signal are caused by the dispersion penalty. Figure 4b, e shows the differential outputs of the TSADC system after incorporating the FEOM. The complementary fading characteristics between the $P^{(+)}$ and $P^{(-)}$ outputs are evident. Figure 4c, f depicts the response of the system after applying the MRC algorithm. As can be seen, the original signal is fully recovered. The small amount of distortion in the recovered signal is due to undersampling in the simulations.

## Application in time-stretch instruments

The modulator was designed to possess an EO bandwidth high enough to effectively capture the first few nulls of the $P^{(+)}$ and $P^{(-)}$ outputs in the frequency response, as illustrated in Fig. 4e. This was to allow for a clear observation of the complementary fading characteristics of the two outputs. The 3-dB bandwidth of the fabricated FEOM was determined and resulted in an estimated value of approximately 44 GHz. Further, the low-frequency half-wave voltage, $V_\pi$, of the devices was measured to be 7.66 V for a modulation length of 0.7 cm, resulting in a $V_\pi L$ of 5.36 V cm. It is worth highlighting that if the current devices were configured in a standard push-pull configuration, as commonly pursued in the TFLN modulator literature[5], the measured $V_\pi L$ would be halved to 2.68 V cm.

The functionality of TFLN FEOMs is rigorously confirmed through experimental verification. The setup is based on a time-stretch enhanced recording (TiSER) oscilloscope, which is the single channel version of a TSADC, as illustrated in Fig. 5a and elaborated in the Methods section. The oscilloscope's sampling rate is set at 50 GSa s$^{-1}$ and the stretch factor of the system is 9.2, resulting in an effective sampling rate ($f_s$) of about 460 GSa s$^{-1}$ for the TSADC. The total effective jitter is another important performance parameter of TSADC systems, which is calculated as[2]

$$\tau_{j,\text{eff}} = \sqrt{\tau_{j,\text{laser}}^2 + \left(\frac{\tau_{j,\text{clock}}}{S}\right)^2}, \qquad (3)$$

where $\tau_{j,\text{laser}}$ is the inter-pulse jitter of the laser and $\tau_{j,\text{clock}}$ is the clock jitter of the digitizer. The digitizer implemented in the present study featured an rms sampling jitter of 270 fs. The use of a single-shot system, such as TiSER, effectively negated any timing jitter that may have been present in the mode-locked laser. As a result, the effective jitter of the TSADC is ~29.4 fs.

After preliminary characterization of the FEOM and TSADC system, the modulator's differential and phase diversity capabilities were examined using the measurement setup (Fig. 5a). During the measurement, the FEOM is biased at its quadrature point to eliminate the second-order intermodulation distortion[65], and fed by a signal generator. To minimize the effect of third-order distortions, it is ensured that the modulator is not overdriven. The normalized RF transfer functions of the $P^{(+)}$ and $P^{(-)}$ channels, after performing differential and phase diversity operations, are shown in Fig. 5b. The MRC algorithm was used to exclude the effect of the dispersion penalty and retrieve the original signal, which was performed digitally on the $P^{(+)}$ and $P^{(-)}$ branches. The first nulls in the frequency response appeared at ~13.3 GHz and ~27.5 GHz, which were in general agreement with the simulation results presented in Fig. 4e. Frequency roll-offs are evident

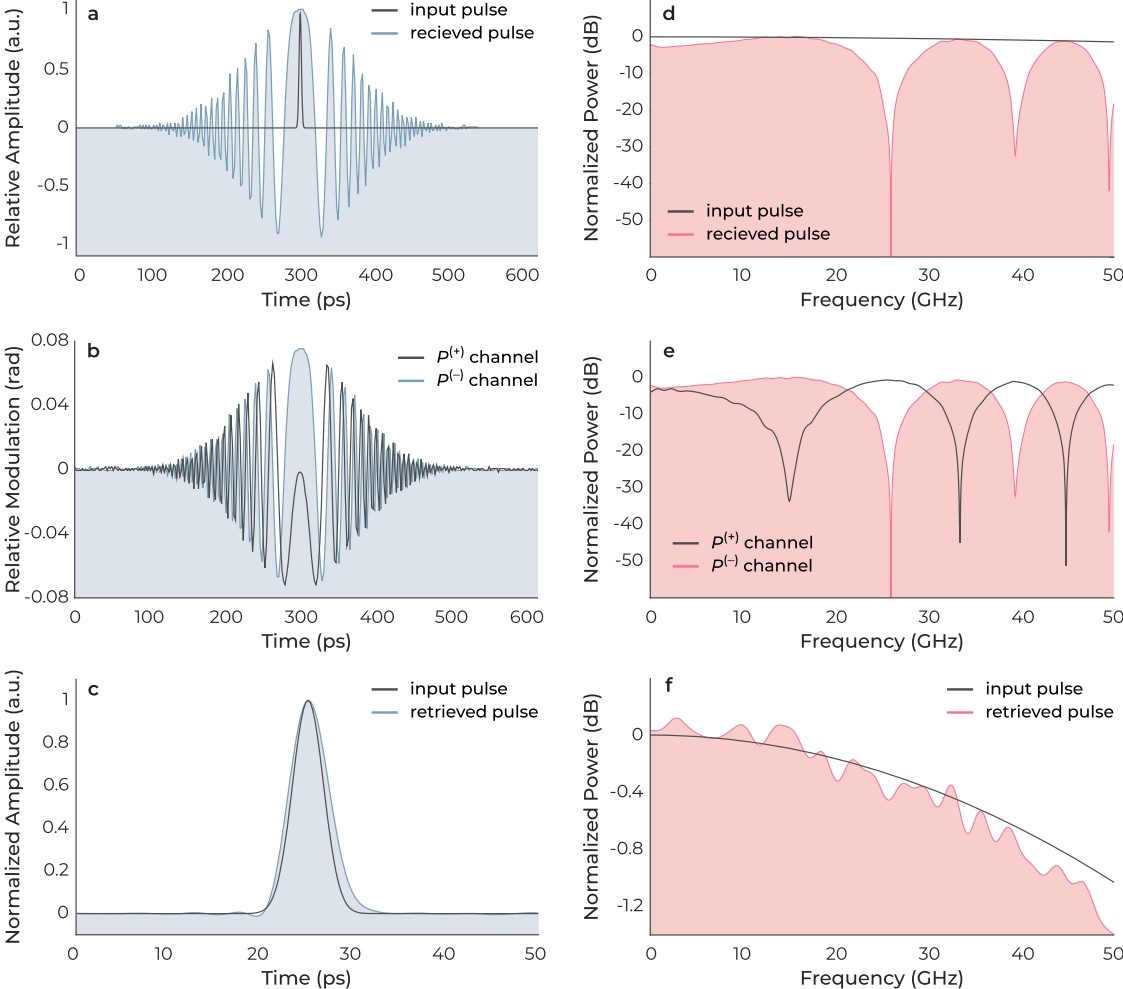

**Fig. 4 | Concurrent employment of differential and phase diversity in a four-phase electrooptic modulator (FEOM).** The time-domain representation of an input and a retrieved pulse in a time-stretch system **a** Without incorporating FEOM. **b** After utilizing FEOM. **c** After performing the maximal ratio combining algorithm. **d–f** Exhibits the frequency domain response of the same pulses shown in **a**–**c**.

in the shown responses. In general, time stretch systems do not inherently introduce any roll-off effect in the measured transfer function. However, both the RF sweep generator and the EOM can cause roll-offs. Furthermore, the FEOM response rolls off due to an increase in the $V_\pi$ as the frequency increases, while the device is always biased at the measured low-frequency bias in the experiment.

## Application in optical communication

Combining the four outputs of the structure in Fig. 1c is not suitable for taking advantage of the differential operation for telecommunication applications. However, if the $\{I, Q\}$ and $\{\bar{I}, \bar{Q}\}$ sets are transmitted through the orthogonal polarization modes of a fiber link, all the information required for differential operation during balanced detection can be retrieved. The FEOM in Fig. 1d can achieve this goal. In this proposed device, the $I$ and $Q$ components are multiplexed into one guided-mode polarization and the $\bar{I}$ and $\bar{Q}$ components are multiplexed into another. This polarization-based separation of the four components allows for eliminating the common mode noise using the differential operation after a long-fiber dispersive element.

Simulation of such an FEOM was carried out using a commercial software package from VPIphotonics Inc. The simulation, detailed in the Methods section, endeavored to faithfully replicate a practical communication system by incorporating all relevant physical parameters. Examples are: (1) RIN and (2) linewidth of the transmission laser; (3) the transition type of the pattern generator; (4)

thermal and shot noise of the photodetector; (5) amplified spontaneous emission (ASE) noise of the optical amplifier, including ASE-ASE and signal-ASE beat noises; and (6) loss of the modulator. The output of the FEOM was sent through a 100-km long optical fiber with a dispersion of 16 ps nm$^{-1}$ km$^{-1}$ and demultiplexed before being sent to a pair of optical coherent detectors. The $P^{(+)}$ and $P^{(-)}$ signals were generated by a pair of differential detectors. The $P^{(+)}$ and $P^{(-)}$ signals are depicted in the time and frequency domains in Fig. 6a, b, respectively. The phase-diversity characteristics of the FEOM between the two channels are evident in Fig. 6b. The comparison of the original input signal and the recovered signal, after performing the MRC algorithm, is shown in Fig. 6c, d in the time and frequency domains, respectively.

Figure 6e displays the eye diagram, and Fig. 6f exhibits the associated BER alongside the $Q$ factor, related to SNR, depicting how they are influenced by the laser power at the transmitter. These illustrations offer critical insights into evaluating the performance of optical communication systems. The eye diagram reveals distinct signal quality, and at laser powers higher than 26 mW, the BER falls below the minimum acceptable threshold of $1 \times 10^{-9}$ (corresponding to $Q = 6$[1]). The results in Fig. 6 confirm that the FEOM can be exploited in optical communication systems, as it adeptly eliminates common mode noise and dispersion. Moreover, dispersion compensating fibers (DCF) typically have propagation losses of ~0.6 dB km$^{-1}$ (e.g., YOFC BD NDCF-G.652C/250), which is over three times higher than that of standard

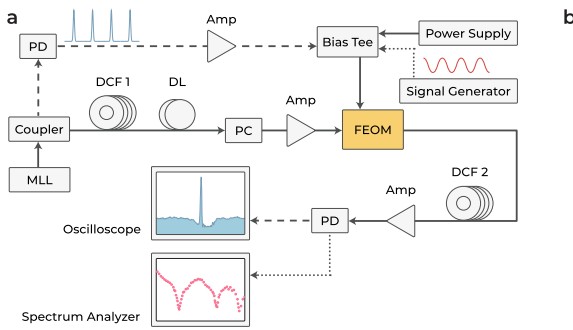

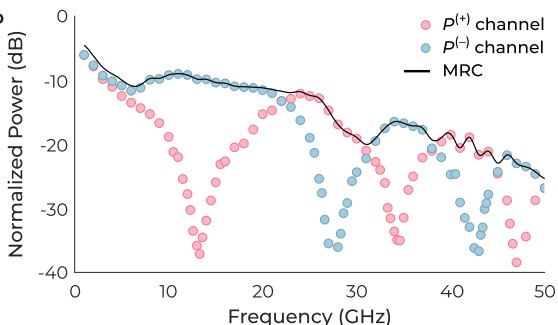

**Fig. 5 | Phase diversity measurement. a** Schematic of the time-stretch enhanced recording (TiSER) system used to examine the differential and phase diversity capabilities of the four-phase electrooptic modulator (FEOM). The temporal domain measurement is represented by black dashed lines, the frequency domain measurement is indicated by black dotted lines, and black solid lines are utilized for both temporal and frequency domain measurements. PD photodetector, PC polarization controller, DL optical delay line, Amp amplifier. **b** The measured transfer functions of the two FEOM outputs in a TiSER system and application of the maximal ratio combining (MRC) algorithm on them.

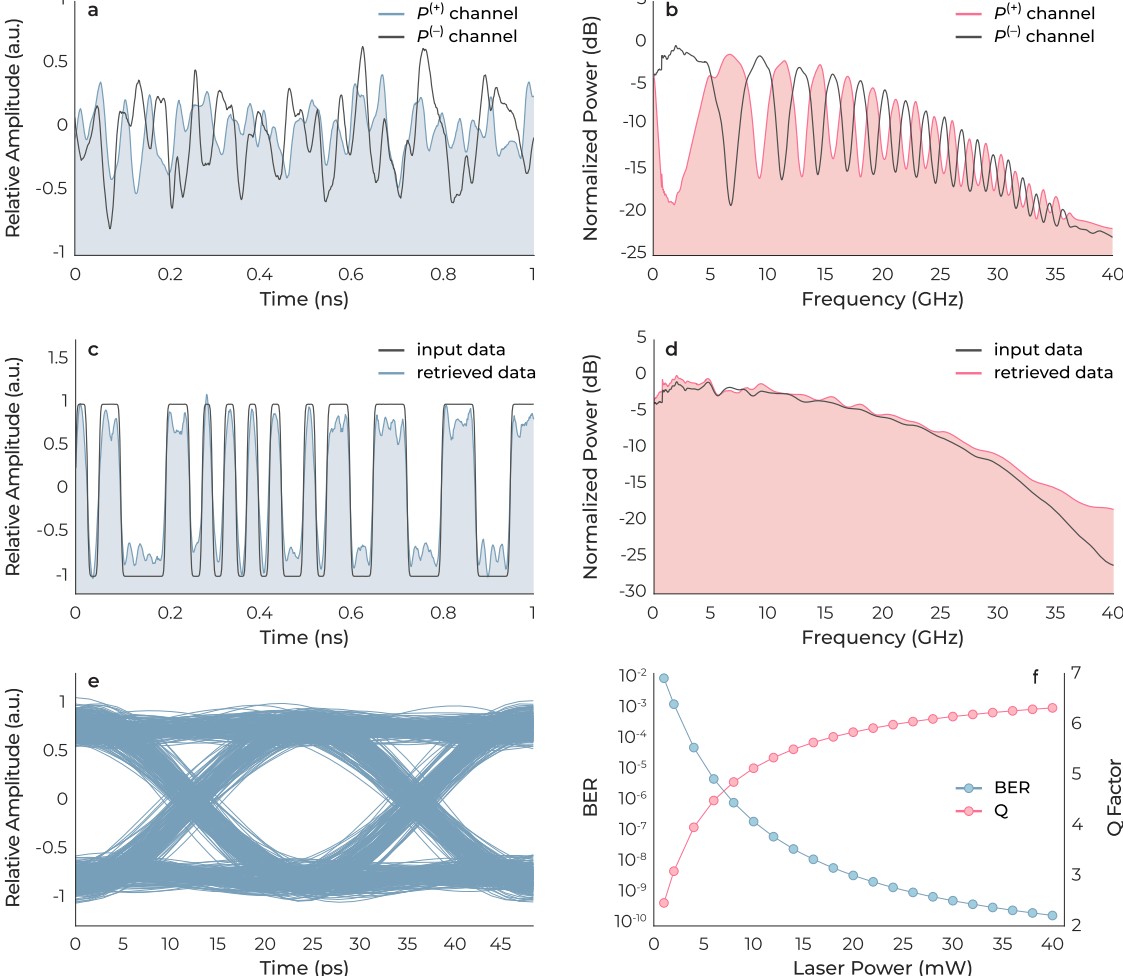

**Fig. 6 | Mitigation of the dispersion effect in optical communication systems. a, b** Outputs of the four-phase electrooptic modulator (FEOM) communication link after performing differential and phase diversity operations in time and frequency domains, before performing maximal ratio combining (MRC) algorithm. **c, d** Display the same results as **a** and **b**, but only after performing the MRC algorithm. For clarity, the input data is also included. **e** The eye diagram of the retrieved signal. **f** The BER and Q factor of the recovered signal as functions of the laser power at the transmitter.

G.652 single-mode fibers (SMFs). Eliminating DCFs can hence improve the power budget of communication links.

One downside of the proposed FEOM is that it has only one data input port, as opposed to two in QPSK modulator, or four in a DP QPSK modulator (Fig. 1). In other words, the bit per symbol of an FEOM is two

to four times less. This is because the two polarizations are utilized to obtain phase diversity and differential channels in FEOMs. However, by eliminating the dispersion penalty of the fiber, the reduced number of data inputs in FEOMs is counteracted by the significantly longer transmission distance.

## Discussion

A class of integrated photonic devices, namely FEOMs, has been proposed, fabricated, and characterized. The architecture effectively surmounts the bandwidth and dynamic-range limitations of photonic systems due to dispersion penalty and semiconductor laser noise, respectively. The architecture enables the concurrent execution of phase diversity and differential operations on a single PIC and is implemented on the TFLN platform. The circuit comprises two nested MZMs. It has been verified that the proposed FEOM is capable of canceling the dispersion penalty and noise in a dual-polarization optical communication link. Furthermore, the FEOM is augmented by two dispersive optical-fiber elements and fiber-optic delay lines, for time-stretching and synchronization, respectively. It is experimentally demonstrated that the inherent nulls in the frequency response of a TiSER oscilloscope can be eliminated. This demonstration is a significant achievement and a noteworthy advancement in the practical implementation of photonic time-stretch systems and coherent optical communication systems. The proposed modulator is more complicated than a standard MZM, leading to a larger chip size and potentially lower fabrication yield, which will increase the cost. The TFLN platform is ideal for mitigating some of these limitations. We believe that the phase diversity and differential advantages justify the added complexity.

## Methods

### Mathematical framework

A detailed analytical analysis of FEOM is provided here. Here, we use the notation $E_s(t)$ and $\widetilde{E}_s(\omega)$ to denote the electric field in time and frequency domains, respectively. The subscript $s$ corresponds to the steps 1–5 in our experimental setup (Fig. 2).

In the first step, we apply a frequency-dependent phase shift to the output pulses of a supercontinuum source, $E_1(t)$, using GVD in an optical fiber with length $L_1$ and second-order dispersion parameter $\beta_2$. This transformation results in the generation of chirped carrier pulses such that

$$\widetilde{E}_2(\omega) = \widetilde{E}_1(\omega)\, e^{-j\omega^2\beta_2 L_1/2}. \tag{4}$$

Here, we neglect the non-quadratic phase shifts induced by the third-order dispersion parameter. Then, the chirped electric field enters the FEOM, which is composed of four waveguides. The electric fields propagating along the first and the fourth waveguides acquire only the spatial phase due to propagation. However, the electric fields in the second and third waveguides accumulate additional phases of $\phi(t)/2$ and $-\phi(t)/2$, respectively, due to the applied electric field to the coplanar waveguide. Here, $\phi(t) = m\cos(\omega_{RF} t)$ is the modulation phase by the single tone electrical signal of frequency $\omega_{RF}$ and the modulation index $m = \pi V_{amp}/V_\pi$, where $V_{amp}$ and $V_\pi$ represent the signal amplitude and the half-wave voltage of the modulator, respectively. Hereafter, we use $\phi$ instead of $\phi(t)$ to simplify the notation. The output electric field of the modulator in each component $k$ can be expressed as

$$E_3^{(k)}(t) = \frac{\sqrt{2}}{4} E_2(t) f(t;k), \qquad k \in \{I,Q,\overline{Q},\overline{I}\} \tag{5}$$

where $f(t;k) = [1 - j\exp(j\phi/2), \exp(j\phi/2) - j, \exp(-j\phi/2) - j, 1 - j\exp(-j\phi/2)]$, respectively. All components $k$ are labeled in Fig. 2. In the next step, we expand the phase terms $\exp(\pm j\phi/2)$ in a Taylor series and make the linear approximation, ignoring the second and higher order terms of $\phi$. Under this approximation, we write the Fourier-domain representation of the field in equation (5) as

$$\widetilde{E}_3^{(k)}(\omega) = a_1\widetilde{E}_2(\omega) + a_2 m\, e^{j\delta(k)} \left[\widetilde{E}_2(\omega - \omega_{RF}) + \widetilde{E}_2(\omega + \omega_{RF})\right], \tag{6}$$

where $\delta(k) = [\pi/4, 3\pi/4, -\pi/4, -3\pi/4]$, respectively, and terms $a_1 = (1/2)\exp(-j\pi/4)$ and $a_2 = (1/8\sqrt{2})\exp(-j\pi/4)$ are constant complex coefficients. Propagating through the second GVD component of length $L_2$, the electric field becomes

$$\widetilde{E}_4^{(k)}(\omega) = \widetilde{E}_3^{(k)}(\omega)\, e^{-j\omega^2\beta_2 L_2/2}. \tag{7}$$

Using equations ((4)–(7)), we can write $\widetilde{E}_4^{(k)}(\omega)$ as a function of $\widetilde{E}_1(\omega)$, where the terms $\omega \pm \omega_{RF}$ appear. For wideband supercontinuum pulses with slow frequency-dependent variations ($\Delta\omega_{optical} \gg \Delta\omega_{RF}$), we can use the approximation $\widetilde{E}_1(\omega \pm \omega_{RF}) \cong \widetilde{E}_1(\omega \pm \omega_{RF}/S)$, where $S = 1 + L_2/L_1$ denotes the time stretch factor. Using this approximation, the Fourier-domain electric field can be summarized as

$$\widetilde{E}_4^{(k)}(\omega) = \widetilde{E}_{env}(\omega) + \left(\frac{a_2}{a_1}\right) m\, e^{-j(\phi_{DIP} - \delta(k))}\left[\widetilde{E}_{env}\left(\omega - \frac{\omega_{RF}}{S}\right) + \widetilde{E}_{env}\left(\omega + \frac{\omega_{RF}}{S}\right)\right], \tag{8}$$

where $\widetilde{E}_{env}(\omega) = a_1\widetilde{E}_1(\omega)\exp(-j\omega^2\beta_2 L/2)$ shows the envelope function of the electric field, $\phi_{DIP} = \omega_{RF}^2\beta_2 L_2/2S$ is the dispersion phase, and $L = L_1 + L_2$ is the total length of the GVD elements. By applying the inverse Fourier transform to equation (8), the time-domain electrical field will be

$$E_4^{(k)}(t) = E_{env}(t)\left[1 + \left(\frac{a_2}{a_1}\right) m\, e^{-j(\phi_{DIP} - \delta(k))}\left(e^{j\omega_{RF}t/S} + e^{-j\omega_{RF}t/S}\right)\right]. \tag{9}$$

We calculate the output photocurrents of the components from $P(t) = (c\epsilon_0 n\eta A_{eff}/2)E(t)E^*(t)$ where parameters $n$, $\eta$, and $A_{eff}$ denote the refractive index of the fiber, photodetector responsivity, and effective optical field mode area in the fiber, respectively. The photocurrent at each channel can be calculated as

$$P_4^{(k)}(t) = P_{env}(t)\left[1 + \frac{1}{\sqrt{2}}m\cos\left(\frac{\omega_{RF}t}{S}\right)\cos(\phi_{DIP} - \delta(k)) + \frac{m^2}{8}\cos^2\left(\frac{\omega_{RF}t}{S}\right)\right], \tag{10}$$

where $P_{env}(t) = (c\epsilon_0 n\eta A_{eff}/2)E_{env}(t)E_{env}^*(t)$ represents the current in the absence of the modulating electric signal. After performing differential at the BPDs, one obtains equation (2).

### Device fabrication

Low-loss waveguides on 400-nm-thick X-cut TFLN dies are formed using electron-beam lithography (EBL), ZEP520A electron-beam resist, and an inductively coupled plasma etching system. The waveguides, with 110-nm-thick ribs, are then passivated with a 500-nm-thick silicon oxide layer created through plasma-enhanced chemical vapor deposition. After passivation, trenches are created within the oxide by using EBL and reactive ion etching to make space for the formation of RF CPWs. An additional step of EBL is performed, followed by the deposition of a 500-nm-thick gold layer via electron-beam physical vapor deposition. The CPWs are then patterned using the liftoff process. In the final step of the fabrication process, the previous step is repeated to achieve CPWs with a total thickness of 1.0 μm.

### Measurement setup

The optical source is a custom-made supercontinuum mode-locked laser at the center wavelength of 1560 nm, with a pulse width of 500 fs and a repetition rate of 37 MHz. A schematic of the experimental setup is provided in Fig. 5a. The laser pulse is chirped with a dispersion compensating fiber (DCF 1) with $D_1 = -120$ ps nm$^{-1}$ km$^{-1}$. The chirped pulse then passes through a variable delay line (General Photonics, VDL-001-15-60-SS) and a polarization controller (PC). To compensate for the power loss during the coupling of the modulator, the pulse is amplified by an Erbium-doped fiber amplifier (EDFA, Pritel FA-15-L). The RF signal is introduced to the amplified laser pulse at the

fabricated modulator via an RF probe in a ground-signal-ground (GSG) configuration with a bandwidth of 50 GHz. To eliminate the potential for back-reflected signals, a second RF probe in a GSG configuration is utilized to terminate the transmission line with a load impedance of 50 Ω. A bias tee (INMET 64671) is used to supply both DC bias (GW GPC-1850D power supply) and phase modulation using a signal generator (HP 83650B) to the FEOM. The modulated pulse is time-stretched by the second dispersion element (DCF 2), with $D_2 = -984$ ps nm$^{-1}$ km$^{-1}$. Since the dispersion attenuates the laser peak power, another EDFA (IPG Photonics EAD-200-CL) is used to amplify the pulse. Then, a wavelength division multiplexer (WDM) is used to filter out the redundant wavelength. The filter is centered at 1570 nm with an optical bandwidth of 20 nm. Finally, the pulse is detected using a photodetector (New Focus 1554-B) and sent to an oscilloscope or an RF spectrum analyzer.

For differential detection, a 95/5 coupler sends 5% of the optical power into a photodetector (Discovery DSC-30S, 20 GHz) for generating a synchronized RF pulse. The RF signal is amplified with an electronic amplifier (Amp, Multilink MTC5515, 10 GHz) before modulating the chirped laser. The optical delay line (DL) is tuned such that the RF pulse is synchronized with the optical pulse. The final photo-detected signal is digitized using an oscilloscope (Tektronix DPO71604, 16 GHz, 50 GS s$^{-1}$) for time-domain measurements. We measure all four ports of the modulator one by one and perform differential detection digitally (mathematical subtraction). The oscilloscope is set under average mode (average every 16 samples) to reduce the detection noise.

For measuring the dispersion penalty, the coupler after the laser source is replaced by a single-mode fiber. Additionally, the RF signal is a sinusoidal wave from the signal generator. In this experiment, the delay line is not tuned since the relative delay between the RF signal and the optical pulse is no longer relevant. The final output of the photodetector is sent to an RF spectrum analyzer (HP 8592B) to measure the frequency response of the system.

### Simulation of four-phase electrooptic modulator for optical communication

A continuous wave (CW) laser with a power of 20 mW and a RIN of −160 dBc Hz$^{-1}$ was employed as the optical input to the FEOM, which was operating with a modulation index of 0.41 and an insertion loss of 15 dB. A pulse pattern generator transmitted a 40 Gbit s$^{-1}$ pseudorandom binary sequence signal which modulated the optical input. The modulated output, in orthogonal polarization states, traversed a 100-km G.652 SMF with a characteristic dispersion of 16 ps nm$^{-1}$ km$^{-1}$, a loss of 0.2 dB km$^{-1}$, and was subsequently demultiplexed in preparation for optical coherent detection. The local oscillator lasers utilized in the simulation had a power of 1 mW and a linewidth of 25 kHz. Signals $P^{(+)}$ and $P^{(-)}$ were generated by a pair of BPDs characterized by a responsivity of 0.9 A W$^{-1}$, a dark current of 1 nA, thermal noise of $2 \times 10^{-11}$ A Hz$^{-0.5}$, and a transimpedance gain of 2 kV A$^{-1}$. The simulation results are depicted in Fig. 6.

### Data availability

The experimental data and primary simulation results generated in this study are deposited in the Zenodo database under the accession code https://doi.org/10.5281/zenodo.8303211[66].

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

## Acknowledgements

The work was partially supported by the United States National Science Foundation (NSF) Industry-University Cooperative Research Center (IUCRC) Program and L3Harris Corporation.

## Author contributions

The design of the modulator was conceptualized by T.J. while he was a visiting scientist at Jalali Lab at UCLA from November 2018 to October 2019. E.O. conducted mathematical modeling, device simulations, and chip design. Fabrication was carried out by E.O. and M.G.V. at UCF. Device testing was completed by E.O. and F.A.J. at Fathpour Lab at UCF. The experimental demonstration of the time-stretch system, utilizing the modulator, was conducted by E.O. and T.Z. at UCLA. Manuscript preparation was contributed to by B.J., S.F., E.O., M.E., and T.Z. The paper was finalized by E.O.

## Competing interests

The authors declare no competing interests.
