## [Peer Review File · Nature Communications]

Differential phase-diversity electrooptic modulator for cancellation of fiber dispersion and laser noiseREVIEWER COMMENTS

Reviewer #1 (Remarks to the Author):

The manuscript reported a four-phase electrooptic modulator, performing differential modulation and phase diversity operations at the same time, which was fabricated in thin-film lithium niobate. The experiment confirmed the performances. The manuscript may be accepted after revision.

- 1 The novelty of this manuscript is not clear. A comparison between this work and existing EO modulator should be discussed in detail.
- 2 The Introduction part should be rewritten, because the current research progress of EO modulator is absent, and the current presentation is misleading and illogical.
- 3 the general presentation of the text, divided into many small paragraphs, is of poor readability. This should be revised.

Reviewer #2 (Remarks to the Author):

The article presents a modulator that can provide both phase diversity as well as differential modulation, concurrently, which can compensate for the dispersion penalty of the fiber, and overcomes the intensity noise and other common mode fluctuations. The manuscript can be accepted after being revised.

1. In order to highlight the article content, limitations of the design of this modulator electrodes would be better to be mentioned.
2. A figure should be placed immediately after the paragraph in which there is the first reference to this figure, e.g., The position of image 2 should be appropriately advanced. Please check and correct other figures.
3. The experiments about long-haul communication need to be supplemented, and whether this modulator will affect other key indicators of the system link while eliminating dispersion and intensity noise needs to be discussed in depth.

Response to Reviewers

Manuscript Title: Differential phase-diversity electrooptic modulator for cancellation of fiber dispersion and laser noise

We thank the reviewers for their time in reviewing our work, insightful questions, and constructive feedback. Our point-by-point responses are given below. The reviewers' comments are in bold font, and our responses are in plain text. In the revised manuscript file, all the changes are highlighted in yellow, including the new references. We would also like to mention that we have slightly modified the title of the paper to better reflect the revisions made based on the reviewers' comments.

Comments

First Reviewer

1. The novelty of this manuscript is not clear. A comparison between this work and existing EO modulator should be discussed in detail.

The reviewer's comment is appreciated. As stated in the original manuscript, the novelty of this work lies in the demonstration of a class of modulators that are capable of operating with phase diversity and common-mode noise cancellation simultaneously (Pages 2 and 3, Lines 92–97, and Pages 3 and 4, Lines 123–169). These operations cannot be achieved in standard or existing configurations. The structure of the proposed modulator is shown in Fig. 1 and compared with the QPSK variation of standard Mach-Zehnder modulators (MZMs). For further comparison with existing EO modulators, the following paragraph has been added to the Introduction on Page 3, Lines 98–114:

“Optical modulators are key devices in any communication or signal processing system. Phase and amplitude modulations are two fundamental mechanisms to attain such devices. The Pockels effect (e.g., in lithium niobate (LiNbO_3 , LN) [5] and polymers [6]) and the free-carrier plasma effect (mainly in silicon [7]) are commonly employed for phase modulation, while the quantum-confined Stark effect (e.g., in silicon-germanium [8] and compound semiconductors [9–12]) is typically used for amplitude modulation. Mach-Zehnder interferometers (MZIs) and microrings [13–15] are usually exploited to convert phase modulation into intensity modulation. Significant advancements in the performance of optical modulators across various material platforms and structures have been reported in recent years. For instance, modulation bandwidths within the subterahertz range are reported in thin-film

lithium niobate (TFLN) MZIs [16, 17]. Additionally, silicon modulators have shown impressive progress, with reported data rates exceeding 100 Gbit s⁻¹ [18–20] and modulation efficiencies of below 1 V cm [21, 22]. As discussed later, TFLN is the chosen platform for this work. In the following, further deliberation on shortcomings of existing EO modulators based on MZIs is presented.”

2. The Introduction part should be rewritten, because the current research progress of EO modulator is absent, and the current presentation is misleading and illogical.

We appreciate the helpful comment. We have rewritten the Introduction, and it now consists of three different sections. The first section (Pages 2 and 3, Lines 60–97) discusses the general features of optical communication systems and their requirements in terms of dispersion penalty and common-noise issues. The second section (Page 3, Lines 98–114) reviews existing EO modulation techniques and platforms, highlighting some state-of-the-art performances. The third section (Pages 3 and 4, Lines 115–183) discusses the shortcomings of existing MZI-based modulators and the need for phase diversity and differential operations, and promotes the employed platform of thin-film lithium niobate.

3. The general presentation of the text, divided into many small paragraph, is poor readability. This should be revised.

We appreciate the reviewer’s feedback on the readability of the manuscript. Understanding that the initial structure with many small paragraphs could have impeded the flow of information, we have made revisions to consolidate relevant content into fewer, more coherent paragraphs. Below is a summary of the specific changes made, where multiple paragraphs have been combined into a single, cohesive unit:

1. Page 2, Lines 60–81: Combined what were previously three paragraphs into one.
2. Pages 2 and 3, Lines 82–97: Integrated what were previously two paragraphs into one.
3. Pages 3 and 4, Lines 129–169: Merged what were previously two paragraphs into one.
4. Page 5, Lines 201–211: Unified what were previously two paragraphs into one.
5. Page 10, Lines 432–449: Combined what were previously three paragraphs into one.
6. Page 11, Lines 463–479: Aggregated what were previously two paragraphs into one.

Second Reviewer

1. In order to highlight the article content, limitations of the design of this modulator electrodes would be better to be mentioned.

We thank the reviewer for an excellent suggestion. The following paragraph has been added to the manuscript on Page 7, Lines 305–317:

“The RF coplanar waveguides (CPWs) are oriented along the y axis of the X-cut TFLN crystal in order to capitalize on the highest EO coefficient, r_{33} , of LN for TE modes. As shown in Fig. 3a, e, bends are added at both ends of the electrodes to facilitate probing and measurement. The bent regions of the electrodes can lead to metallic loss in the optical waveguide underneath. This limitation is mitigated by a thin silicon dioxide buffer layer but comes at the expense of additional processing steps. Moreover, the variation in electrode thickness between the straight and bent sections could result in impedance mismatch and RF reflection at the junction. These factors, along with the RF bending loss, pose potential

challenges to the modulator's bandwidth. Despite these hurdles, the achieved EO bandwidth (presented in the next section) meets the goal of observing the first few nulls of the dispersive fiber frequency response."

2. A figure should be placed immediately after the paragraph in which there is the first reference to this figure, e.g., The position of image 2 should be appropriately advanced. Please check and correct other figures.

We thank the reviewer for the suggestion. Figures have been relocated to appear immediately after their respective references.

3. The experiments about long-haul communication need to be supplemented, and whether this modulator will affect other key indicators of the system link while eliminating dispersion and intensity noise needs to be discussed in depth.

Thanks for the reviewer's comment. We present more detailed results on our modulator in a telecommunications scenario to bring the results closer to a real experiment. To assess the performance under realistic conditions, we utilize the VPIphotonics Inc. design environment, which is the industry standard tool for the design of optical communication links and systems. In the new results added to the paper, all link phenomena have been considered, which are enumerated and expanded upon in the following addition to the manuscript on Page 11, Lines 463–470:

"Simulation of such an FEOM was carried out using a commercial software package from VPIphotonics Inc. The simulation, detailed in the Methods section, endeavored to faithfully replicate a practical communication system by incorporating all relevant physical parameters. These include: (1) RIN and (2) linewidth of the transmission laser; (3) the transition type of the pattern generator; (4) thermal noise of the photodetector; (5) amplified spontaneous emission (ASE) noise of the optical amplifier, including ASE-ASE and signal-ASE beat noises; (6) shot noise; and (7) loss of the modulator."

After incorporating these effects, we obtained new results and replaced Fig. 6a–d with new figures. Two new plots have also been added; specifically, the eye diagram is shown in Fig. 6e, and the BER and Q factor are depicted in Fig. 6f to evaluate the performance for real communication systems. The waveforms and their spectra validate our concept. We have discussed the details of the simulation, including the simulation parameters, in the Methods section. In summary, we have demonstrated that the proposed modulator increases the transmission distance by mitigating dispersion and common mode noises, which allows for a significant extension of the transmission distance. The following paragraph has been added to the manuscript on Page 11, Lines 480–499:

"Fig. 6e displays the eye diagram, and Fig. 6f exhibits the associated BER alongside the Q factor, related to SNR, depicting how they are influenced by the laser power at the transmitter. These illustrations offer critical insights into evaluating the performance of optical communication systems. The eye diagram reveals distinct signal quality, and at laser powers higher than 26 mW, the BER falls below the minimum acceptable threshold of 1×10^{-9} (corresponding to $Q = 6$ [1]). The results in Fig. 6 confirm that the FEOM can be exploited in optical communication systems, as it adeptly eliminates common mode noise and dispersion. Moreover, the use of dispersion compensating fibers (DCF) typically have propagation losses of $\sim 0.6 \text{ dB km}^{-1}$ (e.g., YOFC BD NDCF-G.652C/250), which is over three times

higher than that of standard G.652 single-mode fibers (SMFs). Eliminating DCFs can hence improve the power budget of communication links.

One downside of the proposed FEOM is that it has only one data input port, as opposed to two in QPSK modulator, or four in a DP QPSK modulator (Fig. 1). In other words, the bit per symbol of an FEOM is two to four times less. This is because the two polarizations are utilized to obtain phase diversity and differential channels in FEOMs. However, by eliminating the dispersion penalty of the fiber, the reduced number of data inputs in FEOMs is counteracted by the significantly longer transmission distance.”

Additionally, the following section has been added to the Methods of the manuscript on Page 16, Lines 691–706:

“Simulation of four-phase electrooptic modulator for optical communication

A continuous wave (CW) laser with a power of 20 mW and a RIN of -160 dBc Hz^{-1} was employed as the optical input to the DP FEOM, which was operating with a modulation depth of 0.41 and an insertion loss of 15 dB. A pulse pattern generator transmitted a 40 Gbit s^{-1} pseudorandom binary sequence signal which modulated the optical input. The modulated output, in orthogonal polarization states, traversed a 100-km G.652 SMF with a characteristic dispersion of $16 \text{ ps nm}^{-1} \text{ km}^{-1}$, a loss of 0.2 dB km^{-1} , and was subsequently demultiplexed in preparation for optical coherent detection. The local oscillator lasers utilized in the simulation had a power of 1 mW and a linewidth of 25 kHz. Signals $P^{(+)}$ and $P^{(-)}$ were generated by a pair of balanced photodetectors (BPDs) characterized by a responsivity of 0.9 A W^{-1} , a dark current of 1 nA, thermal noise of $2 \times 10^{-11} \text{ A Hz}^{-0.5}$, and a transimpedance gain of 2 kV A^{-1} . The simulation results are depicted in Fig. 6.”

REVIEWERS' COMMENTS

Reviewer #1 (Remarks to the Author):

the revised manuscript could be accepted.